# Characterizing the Impact of Data-Damaged Models on Generalization Strength in Intrusion Detection

Laurens D'hooge *, Miel Verkerken, Tim Wauters, Filip De Turck and Bruno Volckaert

IDLab-Imec, Department of Information Technology, Ghent University, 9052 Gent, Belgium
* Correspondence: laurens.dhooge@ugent.be

**Abstract:** Generalization is a longstanding assumption in articles concerning network intrusion detection through machine learning. Novel techniques are frequently proposed and validated based on the improvement they attain when classifying one or more of the existing datasets. The necessary follow-up question of whether this increased performance in classification is meaningful outside of the dataset(s) is almost never investigated. This lacuna is in part due to the sparse dataset landscape in network intrusion detection and the complexity of creating new data. The introduction of two recent datasets, namely CIC-IDS2017 and CSE-CIC-IDS2018, opened up the possibility of testing generalization capability within similar academic datasets. This work investigates how well models from different algorithmic families, pretrained on CICIDS2017, are able to classify the samples in CSE-CIC-IDS2018 without retraining. Earlier work has shown how robust these models are to data reduction when classifying state-of-the-art datasets. This work experimentally demonstrates that the implicit assumption that strong generalized performance naturally follows from strong performance on a specific dataset is largely erroneous. The supervised machine learning algorithms suffered flat losses in classification performance ranging from 0 to 50% (depending on the attack class under test). For non-network-centric attack classes, this performance regression is most pronounced, but even the less affected models that classify the network-centric attack classes still show defects. Current implementations of intrusion detection systems (IDSs) with supervised machine learning (ML) as a core building block are thus very likely flawed if they have been validated on the academic datasets, without the consideration for their general performance on other academic or real-world datasets.

**Keywords:** intrusion detection; network security; supervised machine learning; generalization strength; CIC-IDS2017; CSE-CIC-IDS2018

## 1. Introduction

The digital attack surface is expanding. Every day, more devices are connected to the Internet. Malicious actors also have access to this global network and leverage it for nefarious purposes. Identifying and tracking packets or flows on the network that are (part of) a cyberattack is of obvious utility. Researchers have been working on this problem since at least 1985 [1,2]. During that period, network connectivity was not ubiquitous so researchers started their analysis on the hosts under attack, not looking at network traffic. This type of intrusion detection is called host-based intrusion detection (HIDS). As more and more clients became part of networks, it became necessary to add a second branch to intrusion detection. Network-based intrusion detection (NIDS) tries to model the attacks from network traffic.

These models can exist at different levels of abstraction. Deep packet inspection works based on the data encapsulated in the network packets. Packet-level IDS broadens its view by including features extracted from protocol headers and other metadata [3]. Flow-level IDS does not look at individual packets, but treats them as aggregated in flows.

At every level, a further distinction can be made between rule-based systems and anomaly detection systems. Rule-based systems have the advantage of being able to iden-

tify specific intrusion patterns. These methods are built on signature databases. Malicious activity that has been reported on is transformed into a unique signature. Further occurrences of that pattern will be picked up by the system. The biggest downside to this tailored approach is that it is thwarted by alterations to the attack patterns. This has created an arms race between the malicious actors who employ obfuscation and evolutionary strategies to create mismatches with the existing rules and the defense researchers who combat them with novel techniques to generalize the rules [4]. Anomaly detection systems take a different approach. These try to model behavior and report on deviations from normality. This branch is currently the most popular, because it promises models that have a solid general representation and are thus less likely to be fooled by attackers.

### 1.1. Problem Statement

Almost all ML-IDS research is aimed at improving the state-of-the art classification scores on especially crafted, academic datasets. These contributions are easily recognized as improvements if they outperform previous methods [5–9]. However, model evaluation is only performed within the dataset. Models are never exposed to compatible samples from other intrusion detection datasets. This evaluation strategy cannot answer how well these systems would perform when deployed on real networks. This work is a larger-scale continuation of [10] which found generalization issues when exposing CIC-IDS2017 models to CSE-CIC-IDS2018 data (small experiment, few ML methods, did not include all attack classes).

### 1.2. Research Contribution

This work is the first comprehensive test of how well existing machine learning methods are able to learn meaningful representations of network attacks, tested on related academic datasets. Pure classification results obtained in a previous work [11] show that these methods are very capable of classifying the network attacks contained in the individual subsets of CIC-IDS2017. Even more impressive is that these results were stable even when aggressively reducing the amount of data that the learning algorithms had at their disposal. In this article, a fresh set of models is trained on CIC-IDS2017 with the same data reduction methods to verify the earlier results after which the main contribution of this article is presented. The models, pretrained on CIC-IDS2017, are tasked with classifying the new samples of CSE-CIC-IDS2018. In theory, this should go well, because the results within CICIDS2017 are excellent. In reality, it is shown that the models most often do not learn good higher-order representations of attack traffic (classes). In the cases where they do, there are complications that restrict the practical utility of the tested methods.

### 1.3. Article Outline

The experimental design and results for global binary models and attack-class-specific binary models are the main parts of this article. They are described in Sections 2, 4 and 5. The result sections have intermediate conclusions to make the material more accessible. Section 6 largely centers on a simplified view of the results in which only the generalized performance of the best three models per attack class is considered. To conclude, the key observations and contributions are summarized in Section 7.

### 1.4. Related Work

The related work examines the lack of research into model generalization for ML-NIDS from practical, experimental and theoretical perspectives, as shown in Section 1.4.1, Section 1.4.2 and Section 1.4.3, respectively. It also informs the reader of more fundamental critiques of applying machine learning to the intrusion detection problem. Finally, a few noteworthy attempts at solving the generalization issue from the dataset side are highlighted in Section 1.4.4.

1.4.1. Practical: Lack of Interoperable Datasets

A practical reason for the lack of generalization testing in ML-NIDS is the difficulty of obtaining permission to set up capturing experiments on live, corporate, or academic networks. The next-best option is to test between different academic datasets. That too was almost impossible until recently. Few datasets have been created to test intrusion detection systems and typically neither the experiment design nor the feature extraction process are public. This is starting to change, which in large part is thanks to the efforts made by the Canadian Institute for Cybersecurity, operating at the University of New Brunswick. Their data generation experiments have matured and produced two high-quality datasets in 2017 [12] and 2018 [13]. A 2019 [14,15] iteration that specifically focuses on distributed denial of service attacks (DDoS) has just been published.

1.4.2. Experimental: Defining the Scope of Generalization

Publications such as those by Govindarajan et al. [16] and Lu et al. [17] specifically mention improved generalization by employing ensembles of methods and or preprocessing steps (Kuang et al. [18]). Unfortunately, their definition of generalization is too narrow, because they treat it as synonymous with the test set error. Generalization outside of the (often single) dataset on which the proposed methods have been validated is only ever implied.

A recent survey of the proposed deep learning IDSs which specifically selected approaches that mention improved generalization similarly equates generalization with obtaining improved results on a single dataset [19]. The authors of the survey observed three candidate generalization measurements from the literature: model complexity, stability and robustness. Grouped under the umbrella term regularization, several methods are discussed. Some, such as weight decay, dropout, pooling, or weight sharing apply to neural network-based methods, while others such as data augmentation or adversarial training can be applied more broadly. The main concern of the authors is the trial-and-error that is common in deep learning, brought on by the lack of fundamental understanding of why these models outperform. A mention is given to data augmentation as one of the promising routes to increase generalization.

1.4.3. Theoretical and Fundamental Critiques of ML-NIDS

Applying machine learning altogether as a potential solution to intrusion detection has been questioned in the past, most succinctly by the proponents of rule-based systems. The best phrasing of the issue can be read in a landmark article by Sommer et al. [20] stating that: "It is crucial to acknowledge that the nature of the domain is such that one can *always* find schemes that yield marginally better ROC curves than anything else has for a specific given setting. Such results however do not contribute to the progress of the field without any semantic understanding of the gain." Foregoing the operational perspective in favor of slight increases in classification scores on purely academic datasets without insight into what drives the increase is of little utility. Throughout the text, the authors point at the disjoint between the academic community that envisions models that exhibit generality and the functional but highly specialized tooling that is used in real-world settings.

A well-founded but opinionated piece by Gates et al. [21] challenged the paradigm in network anomaly detection by critically examining the underlying assumptions that have been (and still are) relied on. The authors questioned the copying of the requirements and methods put forth by Denning et al. [1], intended for host-based intrusion detection, to network intrusion detection. Three categories covering nine assumptions are discussed. These include issues with the problem domain (network attacks are anomalous, rare and anomalous activity is malicious), the training data (attack-free data are available, simulated data are representative and network traffic is static) and with operational usability (choice of false alarm rate, the definitions of malicious are universal and administrators can interpret anomalies). Based on their challenges to the assumptions, the authors recommend moving away from equating anomalous traffic with malicious traffic, employing hybrid methods

(classification and anomaly detection), community-based sourcing and the labeling of real samples and periodic redefinition of malicious behavior. Some of these points have since been addressed, but the data aspect remains an active issue, which is why recent critiques of the lack of modern, high-quality data are readily available (2015 [22], 2016 [23] 2019 [24]).

### 1.4.4. Reaching Generalization by Augmenting Datasets

A largely theoretical attempt at actually generalizing data for use in signature-based intrusion detection has been described by Li et al. [25]. They propose three tiers to artificially create a more complete input space. The first level (L1) is to generalize the feature ranges for which they propose strategies for both discrete and continuous extension in a realistic manner. This idea is still actively being pursued but with more advanced methods to model the input–output relation (mostly generative adversarial networks (GANs) [26,27]). Generalization testing by augmenting datasets to create new, compatible test sets has also been performed in other areas where machine learning is dominant with surprising results [28].

## 2. Materials and Methods

The methodology section focuses on two aspects: the data on which the models have been trained (Section 2.1) and the training/evaluation procedures themselves (Section 2.2). The evaluation procedure that was developed for this work is new, but it does reuse the data preprocessing and performance measuring components of the training framework [11]. This is intentional, because changing these components or introducing new parts would influence the results. The evaluation code for unseen data sets includes no retraining components. Pretrained models are kept unmodified to evaluate the new samples.

### 2.1. Included Data Sets

The dataset landscape in intrusion detection is sparsely populated. Many papers published today still work with the KDD collection, recorded in 1998 and published in 1999 or its refresh NSL-KDD (2009-revision). Most of the recent innovation is performed by the Canadian Institute for Cybersecurity. After publishing NSL-KDD, researchers at the institute noted that the lack of up-to-date datasets that can be dynamically (re)generated is a serious problem for the research field. The first iteration of a dynamically generated dataset was presented in 2012. ISCXIDS2012 includes baseline traffic that spans multiple protocols (HTTP, SMTP, SSH, etc.). Profiles for the baseline traffic were derived per protocol from real user activity (called B-profiles). Inside a testbed, these profiles can be used to create more benign traffic. In parallel to this, various attacks were performed (M-profiles). Some of these are complex and multi-stage (such as system infiltration), while others are generated by running existing tools (e.g., HTTP-DoS). This controlled separation enables the requirement to machine-label the data. The researchers make the raw PCAP data available as well as CSV files with the processed, labeled samples. This work relies on the two datasets which were built on the ISCXIDS2012 foundation, CIC-IDS2017 (Section 2.1.1) and CSE-CIC-IDS2018 (Section 2.1.2).

### 2.1.1. CIC-IDS2017

The initial experiment was expanded with more protocols (including HTTPS), a greater variety of attacks, more types of clients and larger networks. A new tool to process the PCAP files (CICFlowMeterV3) was also introduced and made open source (https://github.com/ahlashkari/CICFlowMeter, accessed on 7 December 2022). CIC-IDS2017 contains 5 days of traffic, split into seven subsets. The individual subsets contain attacks from different classes spanning DoS, DDoS, port scanning, botnet, infiltration, web attack and brute force traffic [29,30]. Processed CSV file sizes range from 64 to 270 MB. A merged version of all files was created that contains all 2.8 million samples (1.1 GB).

2.1.2. CSE-CIC-IDS2018

The next iteration was published only a year later. CSE-CIC-IDS2018 expands the infrastructure and moves it to Amazon Web Services instead of an on-site experimental setup. It also contains 10 days with samples from the same classes as those present in CIC-IDS2017. A mapping of this restructuring is shown before the attack-specific results in Table 1. Most of the attack scenarios keep using the same tools as those used to generate CIC-IDS2017. The total volume increased drastically with file sizes between 108 and 384 MB. The merged version contains no less than 9.3 million samples (3.5 GB).

Compatible follow-up versions to network intrusion detection datasets are very rare, but they are required to execute the proposed model evaluation strategy. CIC-IDS datasets were chosen for this analysis because they fit the following criteria: they are large-scale, labeled network intrusion detection dataset with compatible feature sets and extracted with the same tooling and with high consistency between the 2017 and 2018 versions (in both attack classes and tools).

**Table 1.** Mapping of the subsets of CIC-IDS2017 to their counterpart in CSE-CIC-IDS2018.

| Attack Class | 2017 | Tools | 2018 | Tools |
|---|---|---|---|---|
| FTP/SSH brute force | 0 | Patator.py (FTP / SSH) | 0 | Patator.py (FTP / SSH) |
| DoS layer-7 | 1 | Slowloris Slowhttptest Hulk Goldeneye | 1 | Slowloris Slowhttptest Hulk Goldeneye |
| Heartbleed | 1 | Heartleech | 2 | Heartleech |
| Web attacks | 2 | Custom Selenium XSS+bruteforce, SQLi vs. DVWA | 5 | same types, tools undocumented |
| Web attacks | 2 | Custom Selenium XSS+bruteforce, SQLi vs. DVWA | 6 | same types, tools undocumented |
| Infiltration | 3 | Metasploit, Dropbox download, cool disk MAC | 7 | Nmap, Dropbox download |
| Infiltration | 3 | Metasploit, Dropbox download, cool disk MAC | 8 | Nmap, Dropbox download |
| Botnet | 4 | ARES | 9 | Zeus, ARES |
| DDoS | 5 | Low Orbit Ion Cannon (LOIC) HTTP | 3 | LOIC HTTP |
| DDoS | 5 | LOIC HTTP | 4 | LOIC-UDP, High Orbit Ion Cannon (HOIC) |
| Port scan | 6 | Various Nmap commands | - | - |

*2.2. Training and Evaluation Procedure*

A small core framework has previously been developed to evaluate IDS datasets. On top of a common core, there are several modifications, all located in the preprocessing steps to accommodate the specifics of the individual data sets. This experiment is supported by a new code that keeps the specific dataset preprocessing code for CIC-IDS2017 and CSE-CIC-IDS2018, followed by new code that channels the unseen samples to the appropriate pretrained models for classification. An overview of the flow of the experiment is given in Figure 1. Classification is performed by the models without any retraining. The collection of classification metrics by which the models' performance is evaluated are standards in data science (i.e., precision, recall, F1-score, balanced accuracy and ROC-AUC). For clarity, most mentions in this article are in terms of precision recall pairs or balanced accuracy. The remainder of this subsection briefly introduces the twelve supervised learners in Section 2.2.1 and the ways in which the difficulty of the classification was increased in Section 2.2.2.

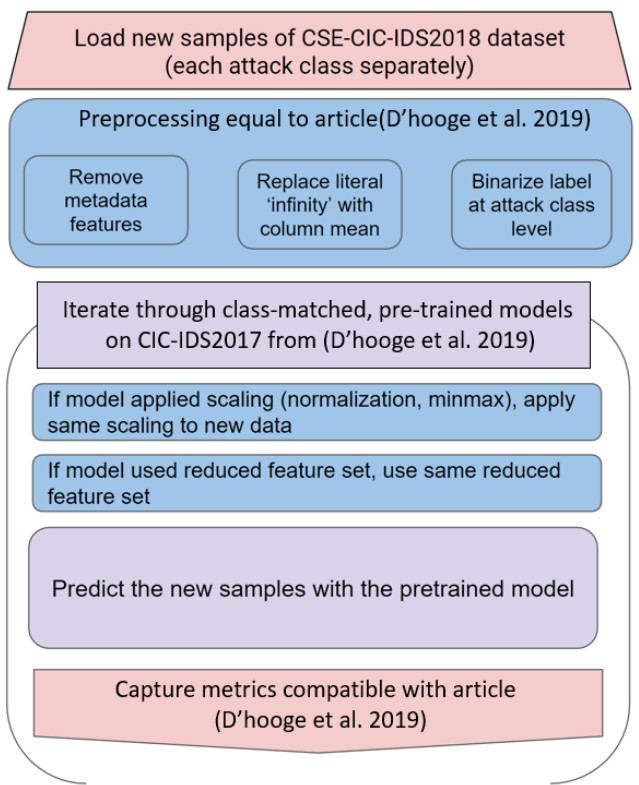

**Figure 1.** A visual overview of the experiment's architecture [11].

2.2.1. Included Algorithms

Pretrained models for a total of twelve supervised classifiers were included. The algorithms are separable into three families. All tree-based methods used gini-impurity to make splitting decisions. The abbreviations next to the methods are used throughout the rest of the text and in the figures. The sequence of decision tree-based classifiers includes important innovations made to them over time. The methods from other families were added for comparative purposes. Detailed information on the inner workings of every algorithm can be read in these references [31–33].

1    Tree-based methods:
- Decision tree (dtree);
- Decision trees with bagging (bag);
- Adaboost (ada);
- Gradient-boosted trees (gradboost);
- Regularized gradient boosting (xgboost);
- Random forest (rforest);
- Randomized decision trees (extratree).

2    Neighbor methods:
- K-nearest-neighbors (knn);
- Nearest-centroid (ncentroid).

3    Other methods:
- Linear kernel SVM (linsvc);
- RBF-kernel SVM (rbfsvc);
- Logistic regression (binlr).

2.2.2. Increasing the Learning Difficulty

The results obtained with the earlier implementation of this framework on NSL-KDD, ISCXIDS2012, CIC-IDS2017 and CSE-CIC-IDS2018, as documented in [11], showed great

classification results. The consistency with which these results occurred in tandem with manual inspection of the intermediate cross-validation results allowed us to conclude that these results are stable and valid. That work already included measures to increase the learning difficulty in an effort to try to invalidate or reinforce the conclusions from a first examination of CICIDS2017 [34].

Data-reduced models are a central component in this work. That reduction was carried out along two axes. The most straightforward of the two is vertical data reduction. This entails reducing the number of samples to learn from through stratified sampling inside a train–test–validation splitter. The models were trained at different points of training volume ranging from (0.1% sample usage to 50% sample usage). The data were first split into a training and test set, with a further split happening on the training set into actual training and validation. Instead of using a fixed portion to test, the complement of the initial split was always taken (e.g., 5% training, 95% test, training further split into training and validation). The second axis is that of horizontal data reduction (i.e., feature reduction). Instead of applying this to strip out the inconsequential features, the opposite was carried out. A list was compiled of the features on which splits were chosen most often in the trees which classified the entire dataset. These top features of CIC-IDS2017 are shown in Table 2. Some features that would obviously contaminate the classification results were removed from the data prior to any training. For CIC-IDS2017/8, these include *Flow ID*, *Source IP*, *Source Port* and *Destination IP*. On a total of 79 remaining features, the 20 most discriminative features were removed before training. This procedure happened in blocks of 5, starting with the best 5 features first, then expanding to remove the 5 next best and so on. Previous findings showed the remarkable resilience of most methods to both horizontal and vertical data reduction [11].

**Table 2.** Most discriminative features of CIC-IDS2017.

| Dataset | | Most Discriminative |
|---|---|---|
| CIC-IDS2017 | 1–5 | Timestamp, Init Win bytes forward, Destination Port, Flow IAT Min, Fwd Packets/s |
| | 5–10 | Fwd Packet Length Std, Avg Fwd Segment Size, Flow Duration, Fwd IAT Min, ECE Flag Count |
| | 10–15 | Fwd IAT Mean, Init Win bytes backward, Bwd Packets/s, Idle Max, Fwd IAT Std |
| | 15–20 | FIN Flag Count, Fwd Header Length, SYN Flag Count, Fwd Packet Length Max, Flow Packets |

## 3. Note on Obtained Results and Graphics

Before presenting the results of this analysis, it needs to be stressed that this article is extensively supported by visualizations to summarize more than 150,000 data points in the result collection. The most interesting results are described in this article, but the total collection is much larger. All visualizations are interactive with the option of changing the parameters and re-render. The result files (grouped in folders D2017-M2017 and D2018-M2017) and associated plotting code are available publicly with documentation on how to run them at https://gitlab.ilabt.imec.be/lpdhooge/reduced-unseen-testing, last accessed on 10 March 2023. In the interest of replication ability, the repository also contains the experiment code required to obtain new results. It is highly recommended to read this article side-by-side with the visualizations.

The results are presented in two separate sections: first the global, binary models' standard intra-dataset performance (Section 4.1) and the same models' performance on unseen, related samples (Section 4.2). Second, because the global, binary models did not remain sufficiently effective, the results of attack-class-specific models are presented in the same way with standard intra-dataset performance first (Section 5.1) and inter-dataset generalized performance second (Section 5.2). Both sections end with brief intermediate conclusions Sections 4.2.4 and 5.2.9.

## 4. Results of Global Two-Class Models

The most hopeful hypothesis is one in which models trained on a large corpus of attack- and baseline traffic would learn an overarching representation between the two classes. This first subsection puts that hypothesis to the test by exposing the models trained on the merged CIC-IDS2017 dataset to itself and then to the merged data of CSE-CIC-IDS2018. The next two subsections delved into the detailed results, while Section 6 offers a summary and short discussion of the best results which is less verbose.

### 4.1. Internal Retest

Retesting the models that have been trained on the merged version of CIC-IDS2017 with their own data shows that these models are consistent with the results described in [11]. This is as expected and it is a necessary requirement to start evaluating the models with samples from CSE-CIC-IDS2018. During the evaluation, five classification metrics were taken into consideration: balanced accuracy, precision, recall, F1-score and standard accuracy.

Every tree-based classifier has classification metrics that converge above 99% with as little as 10% of the data used for training. The neighbor-based methods also stay consistent with previous findings, with knn converging on classification metrics above 98% with 10% of data used fir training. The nearest-centroid classifier fares much worse with a metric profile that is invariant to the amount of data used for training, reaching F1-scores of only 45% (hampered by low precision, and recall is relatively high at 70%). Similarly flat profiles have been observed for the linear support vector machine and the logistic regression. With these models, the F1-score does reach 82%. The RBF-kernel SVM does improve when given access to more training data, reaching an F1-score above 90%.

All models were found to be resistant to feature removal. All tree-based and neighbor methods never lost more than a flat 5% on any metric, even on the most aggressive feature reduction setting, with the removal of the 20 most discriminative features (on a total of 79 available features). The logistic regression and linear SVM did lose up to 10% in flat metrics (i.e., X-10% as opposed to X-(X*10%)). The RBF-kernel SVM never lost more than 5%. Different methods of feature scaling typically had a limited effect on these results. Overall, a case could be made for the normalization of the data over min–max or no scaling, because normalization worked best and most stably for all methods, regardless of algorithmic class.

### 4.2. Exposure to Unseen Data

As stated in Section 2.1.2, CSE-CIC-IDS2018 is very similar to CIC-IDS2017. The 2018 version has the same attacks, executed with the same tools in a different network architecture. One difference is that the 2018 version has a finer division of the attacks, resulting in more dataset fragments (7 in 2017 and 10 in 2018, details in Table 1). This section looks only at the performance of the models trained on the merged version of CIC-IDS2017, tasked with classifying the merged version of CSE-CIC-IDS2018. Based on previous work (summarized in Section 4.1), the expectation is that these pretrained models will work well on the new samples.

#### 4.2.1. Tree-Based Classifiers

Starting with single decision trees immediately shows that the assumption is challenged, because the results are very erratic. The best result is obtained at the 30% training data point, with an F1-score at 63%. Removing the best features incrementally introduces even more variability in the metrics while pushing them downward overall. Using feature removal with min–max-scaling or no scaling at all consistently drops recall below 20%.

A single decision tree was an unlikely candidate to be a good model. Therefore, the analysis included various tree-based ensemble learners. Results for the bagging classifier were not obtained because of insufficient memory on the experiment server (16 GB).

Adaboost had an F1-score at the 30% training data point of 61.0% (recall: 63.0%, precision: 78.3%), close to the performance of the single decision tree. Interestingly, removing the five most discriminative features, improves this point to an F1-score of 65.3% (recall: 71.7%, precision: 82.4%). This lonely peak is gone after removing the top-10 features or more. These observations only exist if the data had been normalized. Min–max or no scaling pushes recall below 20% almost without exception. Precision can be high (80+%) but paired with low recall and thus not useful.

Random forest performs worse, with F1 metric profiles very low (<20%) at almost every point of training volume, especially when applying min–max or no scaling. A singular peak that is similar to the ones for a single decision tree and adaboost happens once more at 30% training volume, but only when removing the top-15 features.

Randomized trees have a worse performance profile than all previous methods with F1-scores stably around 0%, regardless of the training volume or removed features. The "best" result is a meager 10% recall when no scaling is applied, which is invariant to feature removal or training volume. The discrepancy between the performance of this classifier within CIC-IDS2017 and on the related CSE-CIC-IDS2018 is staggering (Figure 2a,b). This method is especially bothersome because of the low time required to build a set of randomized trees. This clearly demonstrates that no learning happens. It is peculiar that a method trained on a thousandth of CIC-IDS2017 (2830 samples) is able to generalize from that to classify the other 99.9% (2,827,913 samples) with an F1-score of more than 98.5%, while completely tanking on the data of CSE-CIC-IDS2018.

Gradient-boosted trees show peaks in recall above 90% when applying normalization or min–max-scaling. One observable pattern from the results is a tendency for these peaks to happen with almost no training data (0.1%). The only downside is the low precision that goes along with the high recall, once again voiding the usefulness of this classifier. Feature removal had inconsistent results for this classifier.

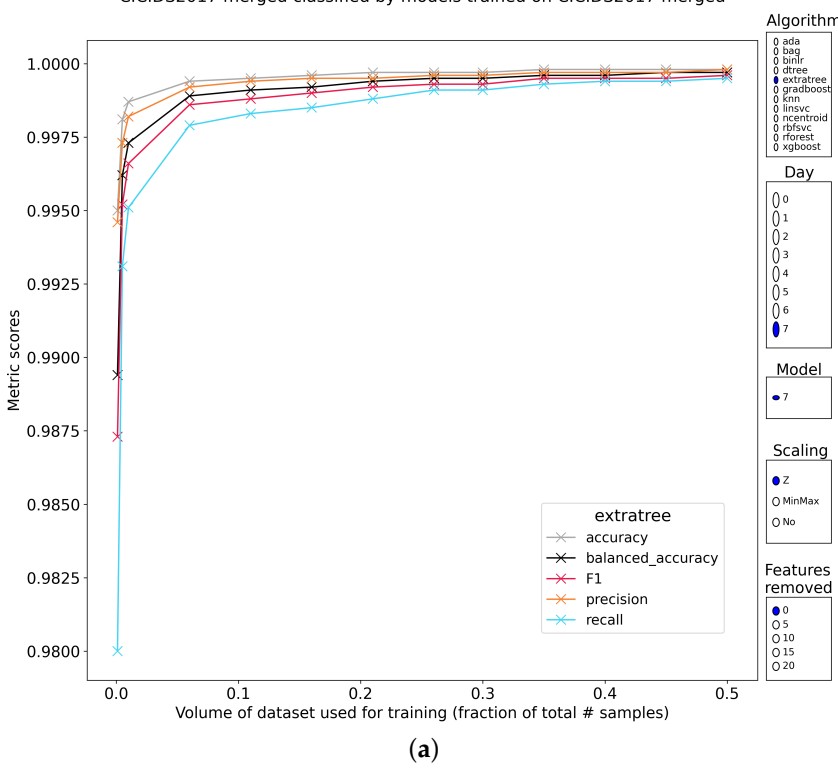

(a)

**Figure 2.** *Cont.*

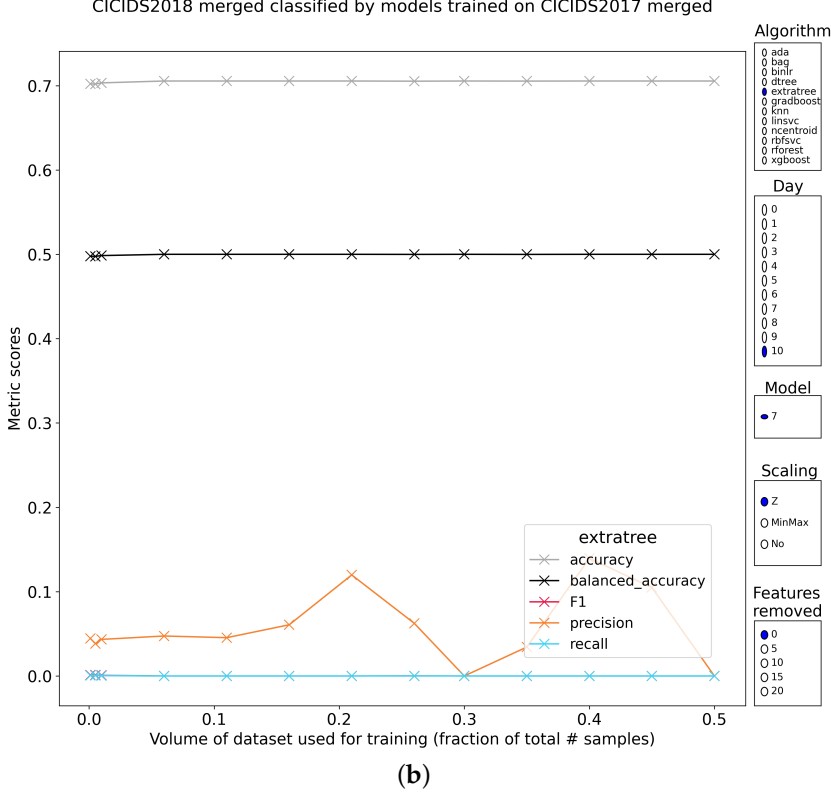

**Figure 2.** Contrast between intra-dataset generalization (**a**); and inter-dataset generalization (**b**) of randomized decision trees on the merged versions of CIC-IDS2017 and CSE-CIC-IDS2018.

Regularized, gradient-boosted trees (XGBoost framework) is the last and most theoretically potent version of a tree-based ensemble classifier in this analysis. Overall, it shows more grouped peaks (decent recall and precision) than the other tree-based classifiers. These results only occur when employing normalization or min–max scaling. The impact of feature reduction is interesting, because the best results are not found when zero feature reduction was applied, but rather when they are found at varying points of top feature removal. The overall conclusion for xgboost remains that it is excessively inconsistent to be usable.

### 4.2.2. Neighbor-Based Classifiers

The simple nearest neighbor algorithm is much more consistent than the tree-based methods. It is only usable when employing normalization, but under that constraint, it reaches F1-scores of approximately 65%. For this algorithm, a clear upward trend in the metrics is observed when increasing the training volume, with diminishing returns starting after 0.5%. Knn is computationally expensive to run, but it can be included in an ensemble, based on these results. Removing the best features in a step-wise manner has the expected result of lowering the classification metrics, but the effect is not drastic and the upward trend stays intact.

The nearest-centroid classifier had the interesting property of having high recall (only for normalized or min–max-scaled features) on CIC-IDS2017. This property stays intact when evaluating the samples of CSE-CIC-IDS2018 with the pretrained models. With normalization and no feature removal, recall stably sits at 70%, as does balanced accuracy (precision 49%). Min–max scaling has even higher recall 87%, but worse balanced accuracy (57.6%, precision 33.7%). Feature removal does not alter the performance when used with normalization, but the models trained on min–max-scaled features significantly improve after removing the first five features. This is most probably due to the removal of the problematic timestamp feature, which was the most discriminative feature in CIC-

IDS2017. Recall now maxes out at 92.8% with balanced accuracy at 57.8% and precision at 33.5%. Removing even more of the most discriminative features does not alter this result. CSE-CIC-IDS2018 has 3.3x the amount of samples that are in CIC-IDS2017 and more importantly, this classifier converges almost immediately (at 0.5% of the samples of CIC-IDS2017 used to train).

### 4.2.3. Other Classifiers

The metrics for a logistic regression show that the features must be normalized to be a decent classifier. It has an upward trend with respect to the training volume, but it is not steep. Generally speaking, this upward trend stays intact when removing features. As expected, the absolute values for the metrics are lowered when reducing features, albeit not by much. At 30% training volume and top-5 features being removed (among others contaminating timestamp being removed) on CIC-IDS2017, it manages to classify the samples of CSE-CIC-IDS2018 with a recall score of 90.3%, precision of 53.8% and balanced accuracy of 77%. The class separation is well above chance, but still not sufficiently high to be able to recommend the method as a reliable classifier.

A support vector machine with a linear kernel has results similar to those of the logistic regression, but there is almost no upward trend and its classification performance is damaged more by feature removal. Its best result is obtained with normalized features, the top-5 of which have been removed and at training volumes of between 0.1 and 1% (recall: 90.5%, precision: 51.5%, balanced accuracy: 77%).

Switching the kernel to the radial basis function has the interesting property of topping out higher, but only for min–max-scaled features. Recall and precision move in opposite directions to one another with regard to the amount of data used for training, regardless of feature removal. With 25% training volume on CIC-IDS2017, recall climbs from 92.2 (top-0 features removed) to 99.45% (top-10 features removed), while precision at the same points drops from 60.5% to 50.3%. A minority of result points have not been collected for this algorithm due to the excessive run time of the algorithm (>1 day per run, caused by the implementation that locks execution to a single core). This classifier could benefit from feature selection in the standard manner (with the removal of poor features instead of the removal of top features). It scores the highest overall, but the time required to train and subsequently evaluate samples holds this algorithm back.

### 4.2.4. Intermediate Conclusion

From these results, it should be clear that generalization is poor at best and dismal at worst. The set of tested algorithm families certainly do not provide a silver bullet algorithm that can be trained to distinguish between benign and malign traffic. Some do have very high recall, but the accompanying precision is lackluster. Tree-based methods have an issue of overfitting despite having great intra-dataset generalization, even under strict limiting conditions. Further research is needed to constrain the tree-based methods to make them more robust. The neighbor-based methods fall into two classes, knn most consistently had the highest F1-scores (between 65 and 70%). Furthermore, it did not require many data points to reach these scores, which is essential for knn, because this is computationally expensive. The method opposite in run time to it, that of nearest centroids, is better in terms of recall and worse in terms of precision. This makes it less usable overall. For the remaining methods, the logistic regression and RBF-kernel SVM have the best results because of their high recall (90–99%), paired with moderate precision (50–60%), but these results are not sufficient to be used in real defense systems. The next section presents the results of testing models specifically trained for each attack class.

## 5. Results of Attack-Specific Two-Class Models

The inability of overarching models to generalize well or at all leads to a new hypothesis in which models trained on specific attack classes may exhibit a better performance. This hypothesis was been tested by tasking the models trained on the individual days

(each containing samples from a distinct attack class) with the data on which they have been trained, as well as the corresponding data from CSE-CIC-IDS2018. This section's subdivisions are rather verbose and therefore quite dense. The summary and discussion of only the best-overall results are in Section 6.

### 5.1. Internal Retest

This section describes the results of making the pretrained models reclassify the samples of the attack class on which they were trained. This is included to test whether the newly trained models do not suffer from a regression in their performance. The classification performance should mirror the results described in this earlier work [11]. The fresh models tested less points of vertical data reduction over a larger range (13 points between 0.1 and 50% of data used for training, versus 35 points between 0.1 and 33%). No other variables were altered in the training methodology.

After comparing the original classification results to the new set of results, no performance regressions were found in the class-specific models. A short reiteration of the results is in order to have a baseline for comparison. CIC-IDS2017 has three classes of attack traffic that were universally well recognized by the tested algorithms.

Models trained on the DDoS, HTTP-DoS and port-scanning traffic subsets are insensitive to reduction in training volume and removal of discriminative features. Put another way: increasing the learning difficulty by scaling back the amount of data for training on while also removing the best features from the data, did not hurt the models' classification scores much or at all. It logically follows that these models are expected to perform well on the samples of these classes from CSE-CIC-IDS2018.

Models trained to recognize FTP/SSH brute force attacks, web attack and botnet traffic are extremely well recognized by tree-based methods, but algorithms from other families have mixed results. Recall tends to stay high, but precision is lost. For all learners, these classes were more sensitive to data reduction, with horizontal data reduction having the biggest negative impact.

The final class, infiltration, is problematic because the subset in CIC-IDS2017 contains a mere 36 positive samples out of a total of 288,602. Results on the CSE-CIC-IDS2018 of these models will be reported, but are unlikely to be good.

### 5.2. Exposure to Unseen Data

The conclusions in Section 5.1 should be promising signs of the good generalization performance of pretrained models when tasked with classifying unseen samples from a closely related dataset. This assumption is only slightly undercut to date by the results from Section 4. Nevertheless, models with increased resolution (i.e., trained to classify only samples from specific attack classes) could perform better. As mentioned in Section 2.1, CIC-IDS2017 and CSE-CIC-IDS2018 are very similar, but the latter has more subsets and volume overall than the former. The mapping between the subsets of these datasets is shown in Table 1.

#### 5.2.1. FTP/SSH Brute Force

Days 0 of both CSE-CIC-IDS2018 and CIC-IDS2017 contain brute force attacks targeted at an FTP or an SSH server. These samples serve as a proxy for brute force traffic in general, because many more service endpoints, both public and non-public exist on the internet that are susceptible to brute force attempts (e.g., API servers, VPN access points, databases, RDP servers, etc.).

The very weak performance for most tree-based models is observed with class separability (balanced accuracy) often at 50%, indicating that the models are no better than chance. The results are worse when employing normalization, because the values are squeezed into a range that is too narrow. Even the models trained on very low training volumes (high vertical data reduction) still overfit on the data. There are some exceptions (subfigures of Figure 3), most notably (and expectedly) when using very low training volumes (0.1%).

Recall is typically very low, but the malicious samples in that recalled percentage are classified with high precision. Models that were trained with the most discriminative features removed fail immediately.

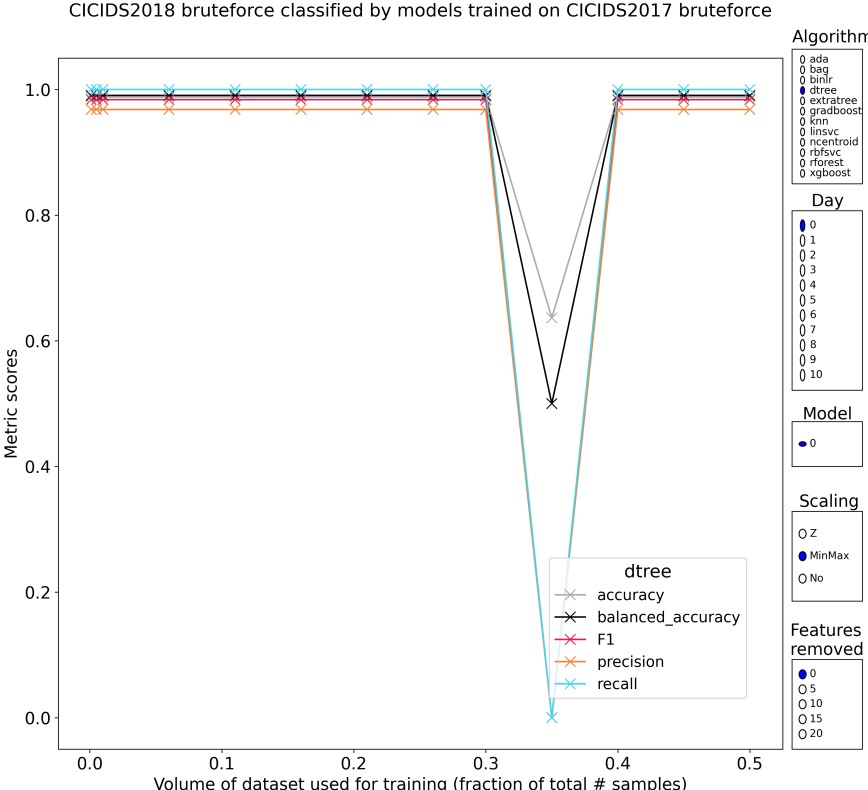

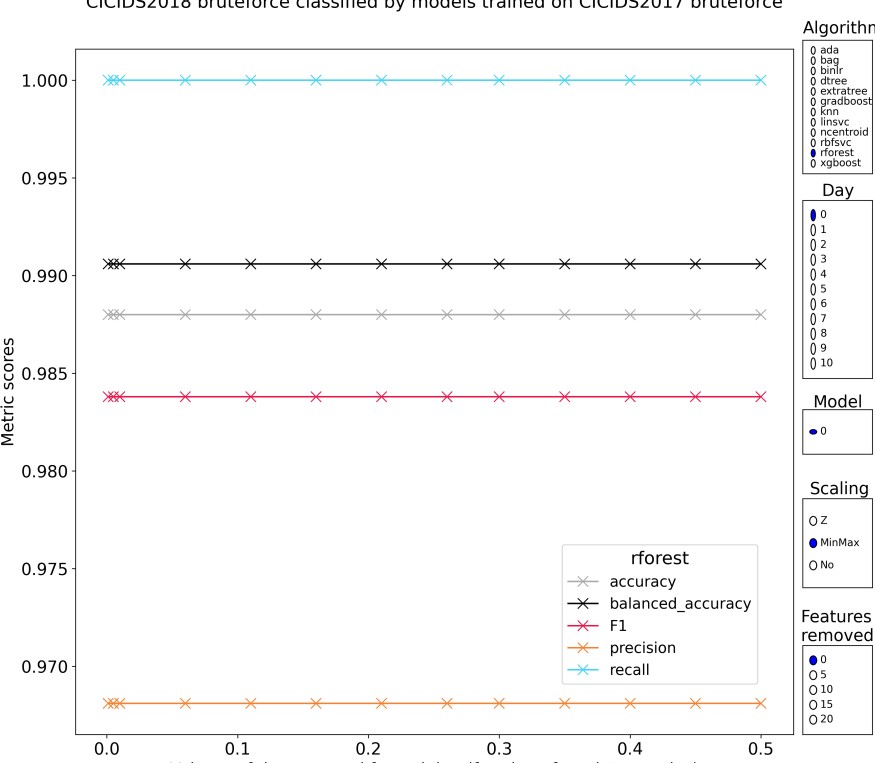

**Figure 3.** Performance metrics of two tree-algorithms trained on FTP-SSH brute force (CIC-IDS2017), evaluating FTP-SSH brute force from CSE-CIC-IDS2018.

A very similar conclusion is reached for the RBF-SVC. It manages to reach moderate recall (71%) with a high precision of 93.8%, but only with normalized features, and 0.1% training volume. Increases in training volume lead to very low recall with high precision. It is more resistant to feature removal than the tree-based methods. The logistic regression and linear SVM do not have noteworthy results.

Nearest neighbors is useless, because it has low precision and recall, regardless of the scaling and invariant to training volume. Nearest centroids has stable sections with precision–recall pairs at approximately 60% and 75%, respectively. These are maintained fairly well when reducing features, but only if the features were normalized or min–max-scaled. Without feature scaling, lots of performance is lost quickly after removing top-features. The classifier seems brittle overall. Curiously, the best recall results, up to perfect recall, happen when using only 0.1% of data for training.

### 5.2.2. Layer-7 Denial of Service

CSE-CIC-IDS2018 contains two days of denial of service attacks. The first of which has malicious packets generated by tools such as slowloris or HULK that abuse web servers by exhausting their resources. The second day contains traffic exclusively from exploiting the Heartbleed vulnerability on the affected implementation of OpenSSL (1.0.1-1.0.1f). CIC-IDS2017 bundles both types of attacks in a single day, using the same tooling. Because the attacks exist in one day in the 2017-version, the attack types got squashed into binary classification. It is a good use-case to test whether these attacks should be treated as the same category or not.

All tree-based methods overfit heavily, as they did on the brute force traffic. Good performance is only ever recorded for models that had very little data available to train on. Making matters worse is the inconsistency with which these results occur. In numerical terms, recall–precision pairs above 60% are very rare for any of the pretrained models. Once more, the worst results are obtained on models that had normalized features.

Nearest-neighbors has balanced accuracy scores consistently falling in the 70–80% range. Changes in scaling, training volume or feature reduction do not significantly alter this result. It is not good enough to be considered. Nearest-centroids separates the classes worse, indicated by the balanced accuracy of 50–60%. The only results that are better than chance were observed when using normalized features. Higher training volume or less feature reduction, do not affect the results.

The logistic regression models trained on min–max-scaled features follow the pattern that the section introduction put forward. Great generalization performance, with a stable, straightforward relationship between training volume and classification metrics (Figure 4). Those metrics are a stable 97.5% recall, paired with 70–75% precision yielding a total of 97.9% balanced accuracy (5 features removed). This amount of class separation is enough to recommend the classifier as a genuine method to classify unseen layer-7 DoS traffic. A linear support vector classifier or rbf-kernel SVM (with the same parameters) have nearly identical results. All models in the other category perform poorly if the features were standardized.

### 5.2.3. DoS Heartbleed

Although technically a form of information disclosure and not denial of service, Heartbleed traffic was included in the DoS category by the authors of both datasets.

Decision trees typically have very erratic metric graphs for this day of traffic. Models at peak performance in these graphs manage to achieve 80–95% balanced accuracy. Adaboost has the highest scores, both in absolute terms as well as averaged across the tested parameters. The changes in classification metrics can be as large as 50% flat and the relationship to training volume and feature removal is unclear. This unpredictability considerably lowers the real applicability of these models. The lowest variability models are randomized decision trees. These reach a flat profile after 1% training volume, with recall at 72.4% and precision at 99.5% (Figure 5). This result is also stable with regard to

feature removal. It should be noted that with 0 features removed (which includes the problematic timestamp feature from CIC-IDS2017), this model performs no better than chance. Performance numbers are only good if this feature was removed ('timestamp' is first in the list of top-5 features).

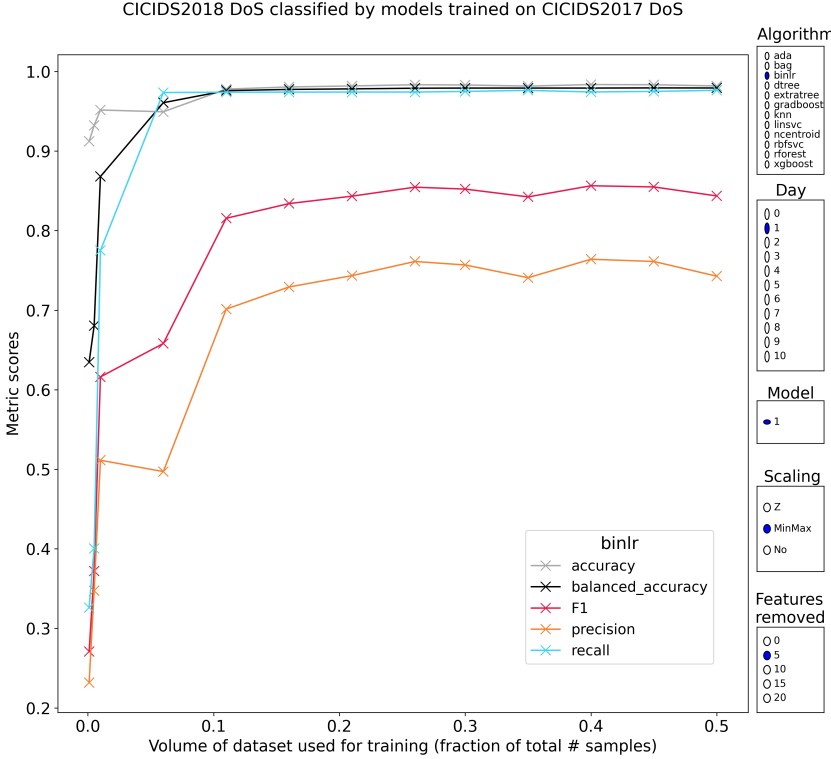

**Figure 4.** A rare occurrence of the expected relation between training volume and generalized model performance.

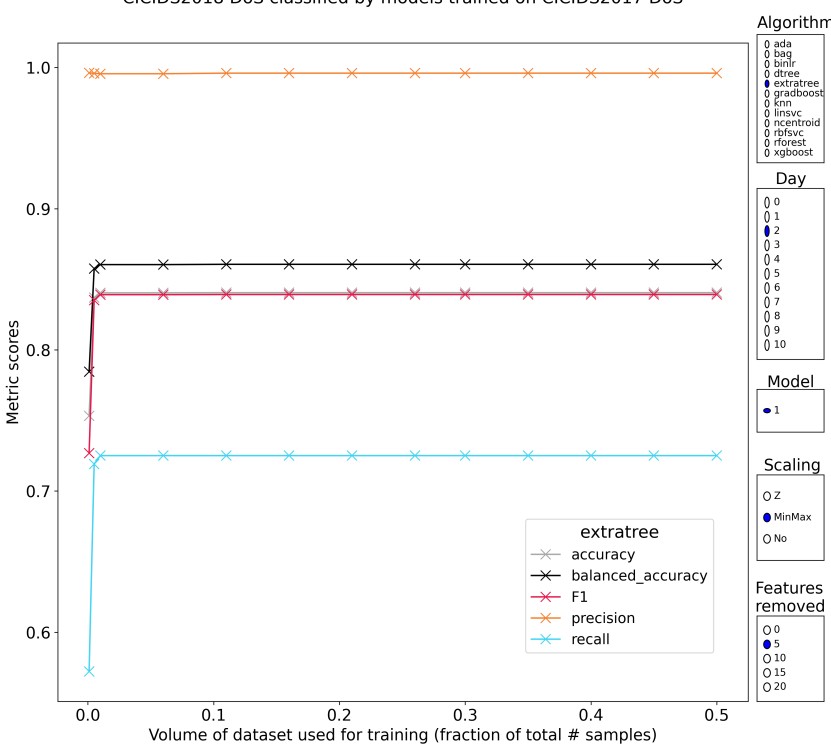

**Figure 5.** A subset of the randomized trees trained to recognize Heartbleed traffic perform stably well.

Nearest neighbors is an unusable classifier. It has very low recall/precision for all methods of feature scaling, across all points of feature reduction. It also shows a sudden decline in performance when using more than 1% of data as training samples. Nearest-centroids had recall–precision pairs of 60 and 95% within the relevant day of CIC-IDS2017, on the day containing Heartbleed samples, the model performance drops to precision recall pairs of 15 and 0% moving balanced accuracy close to blind guessing. This shows just how brittle the classifier is.

The logistic regression was very performant within CIC-IDS2017 with stable metric clusters above 95%, step-wise gain with increased training volume up to limit and step-wise loss in these metrics with increasingly aggressive removal of top features. It has this profile on the new samples of CSE-CIC-IDS2018. The model only starts to become performant with at least 5% data as training volume. The method is stable with perfect precision and 95% recall (Figure 6). This would be usable in a real-world system. Removing features has the expected effect of lowering the overall metrics, but stability is kept. A linear support vector machine has similar results, but requires normalized features. The RBF-kernel SVM required min–max-scaling and feature removal impacted precision much more negatively than it did for binlr.

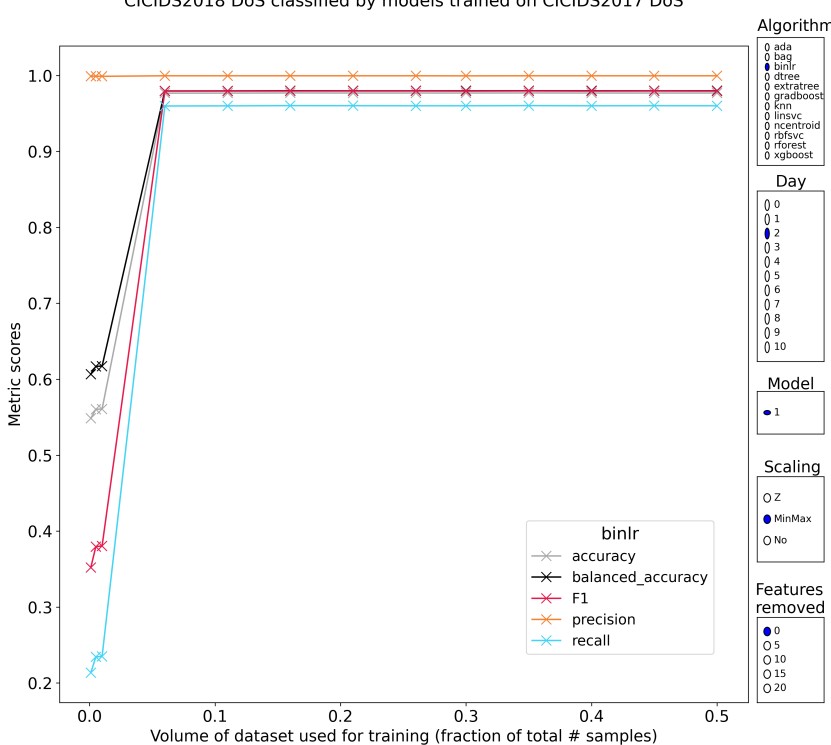

**Figure 6.** A subset of the logistic regression models trained to recognize Heartbleed traffic also have stable, high-performance scores.

Whether it is justified to clump layer-7 DoS and Heartbleed together in CIC-IDS2017 is unclear. The models might be more performant on the individual attacks if they were trained exclusively on them. The argument in favor of keeping the grouping is that there are iterations of the models that manage to classify both types. Testing the exact entanglement could not be deduced from these data, but it is possible by testing models pretrained on CSE-CIC-IDS2018.

### 5.2.4. DDoS Part 1

As with DoS traffic, CSE-CIC-IDS2018 also splits DDoS traffic over two days, whereas CIC-IDS2017 bundled them. The tooling used in both datasets is the same. The first day of DDoS samples in the 2018 version contains traffic generated by the Low-Orbit Ion Cannon (LOIC) tool, with both UDP and HTTP floods. These attacks do not rely on deviant protocol use, but simply overwhelm the web server(s) on the receiving end. The second day uses the High-Orbit Ion Cannon tool which also employs HTTP (GET and POST), as well as LOIC UDP.

All single decision tree models trained on non-scaled features have mirror-image metrics on the DDoS traffic from CIC-IDS2017 and the first day of CSE-CIC-2018 (Figure 7). Adaboost has some models with normalized features that are very performant with tight metric clusters at approximately 97%. Reducing features lowers this performance pulling precision and recall apart to 100% and 80%, respectively. While this could be interpreted as a good result, the unpredictable pattern of these metrics in relationship to the training volume significantly lowers the practical utility of this method. The bagging classifier built on decision trees shows signs of overfitting. It has good stability (normalized features) as long as no more than 10% of the DDoS data in CIC-IDS2017 has been used to train the model. In that low training volume region, the classifier has perfect recall, matched by 80+% precision. Randomized decision trees, random forests and gradient boosted trees, both standard and regularized, do not perform well enough to be considered real contenders.

Lots of tree-based methods show signs of overfitting beyond using more than 5% of data to train on. Methods to improve generalization for tree-based classifiers in intrusion detection are worth investigating.

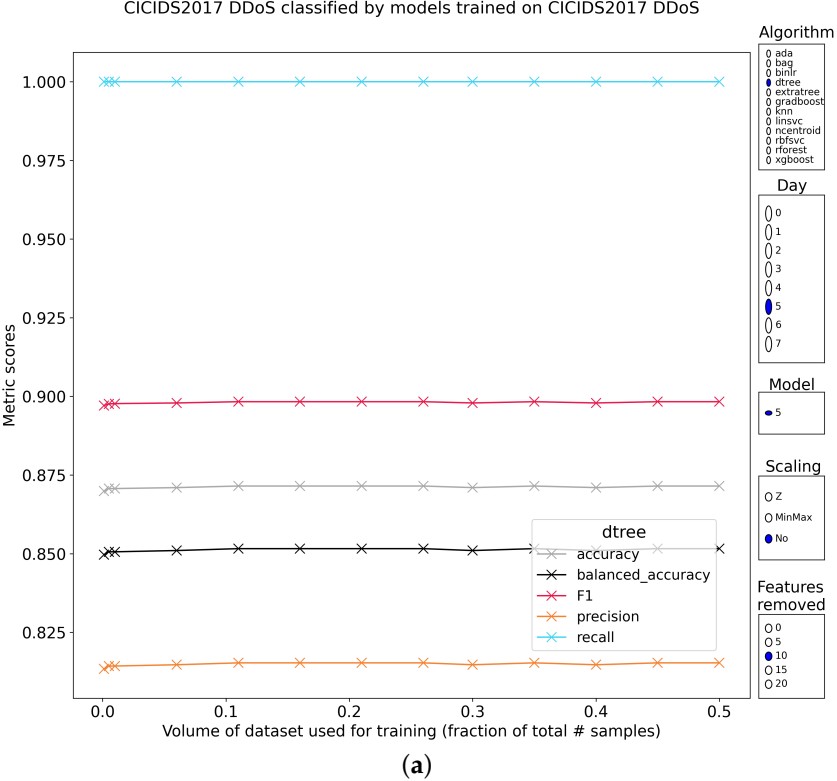

(a)

**Figure 7.** *Cont.*

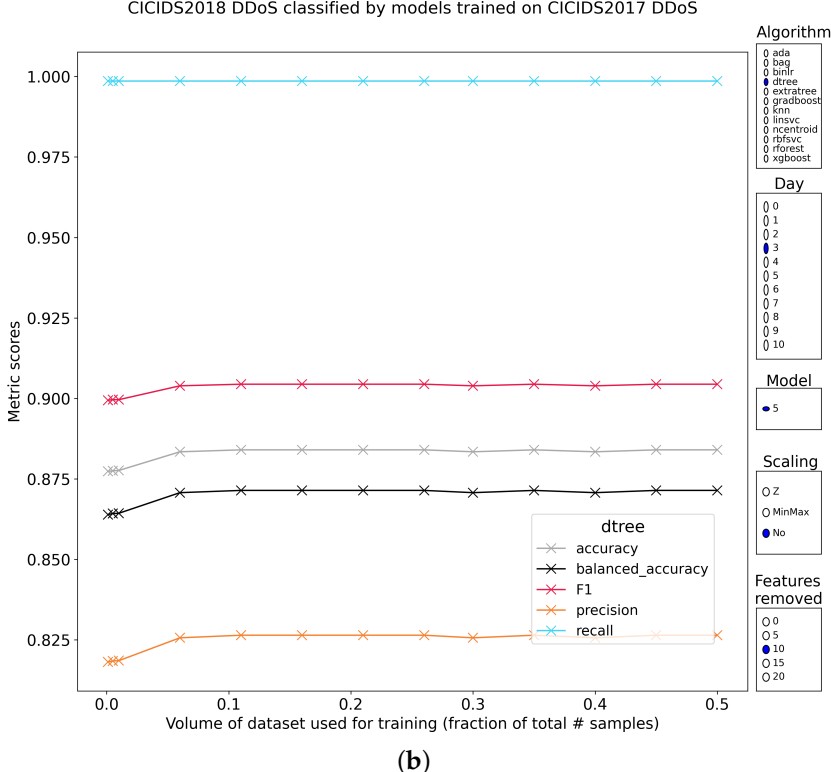

**Figure 7.** A rare occurrence of perfect consistency by pretrained IDS2017 DDoS models the respective IDS2018 DDoS samples. (**a**) Singular decision trees within CIC-IDS2017 DDoS; and (**b**) the same set of models summarized in figure (**a**) when evaluating CSE-CIC-IDS2018 DDoS part 1.

Nearest neighbors with normalized features has perfect recall and reaches 89.5% precision with 10% data used for training. At least a flat 10% loss in precision is observed compared to the classification performance on CIC-IDS2017. Min–max-scaled models also perform well, but not beyond 1% training volume. The nearest centroids loses a flat 10–15% on all metrics compared to the same model's performance on DDoS 2017, but the stability is retained. Recall is poor at only 50%.

The logistic regression models work with little training data reaching perfect recall and 80+% associated precision (only with min–max scaling). Intermittently, there are signs of overfitting at higher training volumes. The linear support vector classifiers obtain equal results, but with less stability. The rbf-kernel SVMs have similar results with both types of scaling. In terms of generalization strength, these methods definitely achieve more stable and thus better results than the other algorithms.

DDoS was one of the easiest classes within CIC-IDS2017. This does not translate into one-to-one to classification strength on CSE-CIC-IDS2018 DDoS. The remarkable resistance to data reduction in all methods of the DDoS class does not hold up. These methods need more robustness to be practical. Evaluated on the whole, classification strength on this easy class is better than it is on the harder classes.

### 5.2.5. DDoS Part 2

The second day of DDoS traffic in CSE-CIC-IDS2018 has very similar traffic. Only one new tool is introduced, and behind the scenes, it generates requests with the same protocol. It is odd to split the DDoS traffic over two days, because as the results will show, performance is alike.

Single decision trees have zones with adequate performance that are interwoven with zones with very poor performance metrics. It is not exclusively due to overfitting either, because regions with good performance do exist at higher training volumes. Adaboost models only work with normalized features, maximally reaching perfect precision and 80%

recall. As a standalone result, this would make adaboost a viable option, but once again, the unpredictability with regard to training volume hampers viability. The pretrained bagging classifiers perform like adaboost at low to very low training volumes and exclusively with normalized features, however, with even less stability. Conclusions for randomized trees, random forests and normal gradient-boosted trees can be summarized as lackluster across all parameters (again with an exception for very low training volumes). Regularized gradient-boosted trees perform stably with 80% recall and 100% precision insensitive to feature reduction.

Nearest neighbors has the same result as in the previous section, but with a worse precision (70+%). HOIC combined with LOIC UDP seems to be harder to classify, because the centroid suffers from very low (<25%) recall compared to the previous section (regardless of parameter selection).

Both methods of feature scaling obtain good results for the logistic regression models, with those trained on min–max-scaled features reaching clusters of perfect metrics. The best scores are obtained at the lowest training volumes, but the differential is tiny in most cases (0.5%). The linear kernel SVM has the same performance profile as binlr, with good results for the models trained on normalized features, but better results on models trained with min–max-scaled features. Once more, the highest performance is obtained with the lowest amounts of training data. The ideal classifier for this attack class is the RBF-kernel SVM with stable, perfect scores. This does require at least 1% data to train on, but shows no signs of overfitting (min–max-scaling).

It could be concluded that ML-based models are able to distinguish well between regular and multiple types of DDoS traffic. Unfortunately, due to the loud nature of DDoS attacks, they are easily detectable by other mechanisms. It might be useful at an aggregate level (service providers), but an individual business suffering from a DDoS attack will not need a machine learning model to confirm that.

### 5.2.6. Web Attacks

The web attacks are a harder attack type to classify within CIC-IDS2017. Most methods were not able to reach perfect classification scores. Although day 5 and 6 contain web attacks, the dataset documentation does not mention what the differences between the two days are. They both contain web brute force attempts, cross-site scripting (XSS) and structured query language injections (SQLis).

The poor generalization obtained by single decision trees is the root cause for the feeble results of the methods that build on top of it. Recall is so consistently below 40%, with spiking precision scores making it impossible to recommend any of these methods. These results did not improve with more training, different scaling or less feature reduction. The worst performers are randomized decision trees. There is no learning, because they do not try to set optimal splitting points. It is clear that for a harder-to-classify attack class, this method does not work. The web attack models built on decision trees typically had 90–100% recall after some training within CIC-IDS2017. The relative 50% drop-off is disconcerting.

Nearest neighbors starts off with some signs of learning, but levels off quickly at low to very low precision–recall pairs. The method had good scores within CIC-IDS2017 (85–95% recall and 75–85% precision), but that performance does not carry over into CSE-CIC-IDS2018. Nearest centroids had robust 85+% recall on the web attack traffic of CIC-IDS2017, at all combinations of training volume, scaling and feature reduction. The associated precision was never good, so it is expected that this will continue. Unfortunately, the nearest centroids loses much in terms of recall. Only the models with min–max-scaled features stay stable at 57%. Other methods of scaling have recall stable at 15%. All precision is lost.

The near-perfect recall and moderate precision of logistic regression models within CIC-IDS2017 is not retained on CSE-CIC-IDS2018. Recall drops below 40%, often crashing to 0%, and precision is at 0% more often than not. This conclusion also applies to a linear

SVM and rbf SVM. All of these methods struggled in terms of precision within CIC-IDS2017, but did reach near-perfect recall. None of this translates into generalization performance when classifying the web attacks in CSE-CIC-IDS2018, despite the fact that both datasets contain the same types of web attacks.

It is clear that, for this type of attack, which typically has a lower network footprint (unless it is a brute force login), is much harder to classify from network-related features. Within the dataset, however, the performance can be very good and a subsequent recommendation for use in real-world systems would be logical, but ultimately misguided.

### 5.2.7. Infiltration

Like web attacks, CSE-CIC-IDS2018 contains two days with infiltration traffic. The documentation does not mention what the differences are between the two days. The labeling in the data does not provide any additional information apart from 'infiltration'. A major caveat when analyzing these results is the lack of samples on which the models were trained. Day 3 of CIC-IDS2017 has infiltration traffic, but the distribution between benign and malicious is extremely skewed (288602-36). Generalization performance is thus not expected.

None of the tested algorithms perform at acceptable rates, and metrics are consistently below 20%. Sometimes, precision spikes high, but the associated recall is so close to zero that the high precision is meaningless. These results do not vary with changes in the training volume, feature scaling choice or feature reduction. The results for both days are close to identical. Some models, especially those built on decision trees, show climbing trends within CIC-IDS2017, but this is just the classifiers fitting any pattern and certainly not one that is significant or general.

It will be interesting to investigate whether models trained on the infiltration days of CSE-CIC-IDS2018 perform well when retested on each other's data as well as the 2017 infiltration samples.

### 5.2.8. Botnet Traffic

The botnet class in CIC-IDS2017 is one of the medium-difficulty classes, mainly because non-tree-based models had low precision and all models suffered from the removal of good features from the training set. The documentation for the 2017 data lists Ares [35] as the tested botnet. The 2018 version adds the Zeus [36] botnet.

Single decision trees have very irregular performance. At some points, the metrics almost reach perfect classification, but it is impossible to reliably tell which parameters are required. Feature reduction tanks performance across all trained models, most notably when using min–max-scaling or no scaling. There is one decent set of models (adaboost) and it requires normalized features and no feature reduction. The resulting models have an early peak at 0.5–1% training volume of perfect recall and 75% precision. Giving access to more training data still yields stable models, but the recall is only 50% with 95+% precision. As soon as features are removed, the classification scores plummet almost to 0. Changing how the features are preprocessed also had a major impact (summarized in Figure 8). The bagging classifier built on decision trees, gradient-boosted trees, random forests and extreme gradient-boosted trees also behave unexpectedly and are not sufficiently potent to be used as a classifier. Randomized decision trees generate no false positives, but no true positives either. All normal traffic is properly classified, but the models miss all malicious instances, leading to a false negative rate of 100% and a total balanced accuracy of 50%. This happens regardless of the training volume, scaling or feature removal (with very few exceptions).

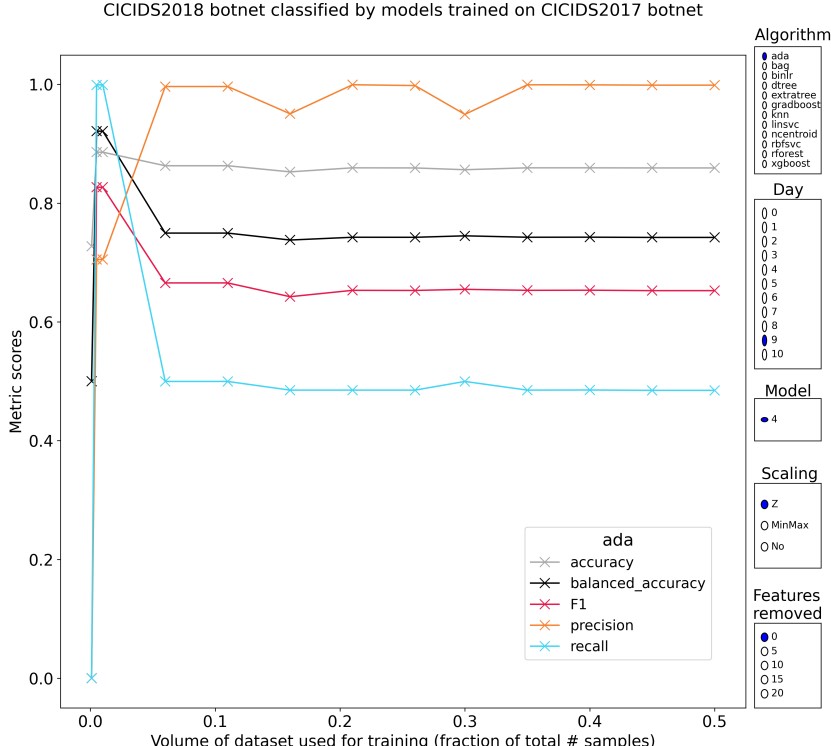

(**a**) Adaboost botnet normalized features, no feature reduction

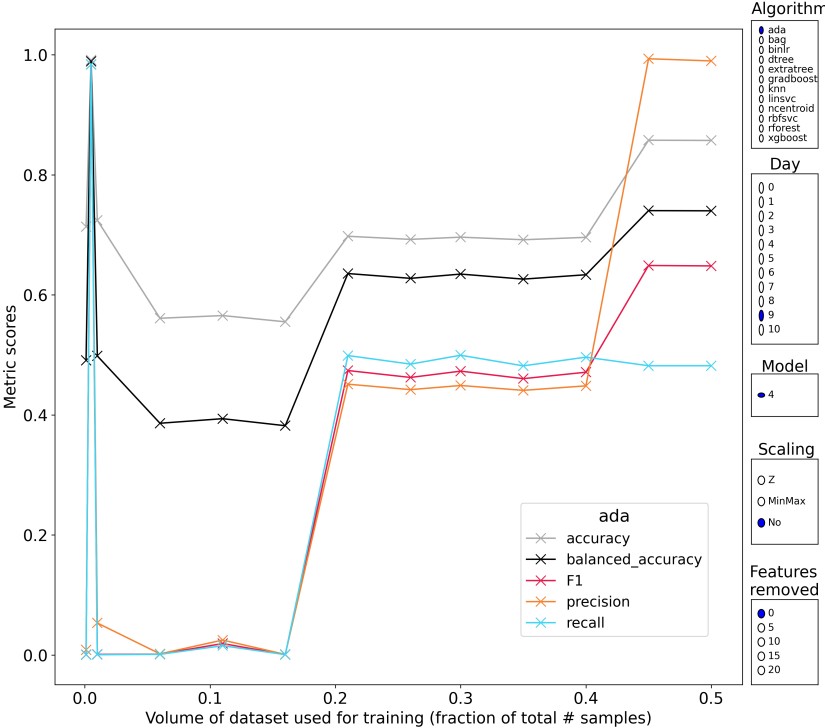

(**b**) Adaboost, pretrained for botnet, no scaling

**Figure 8.** *Cont.*

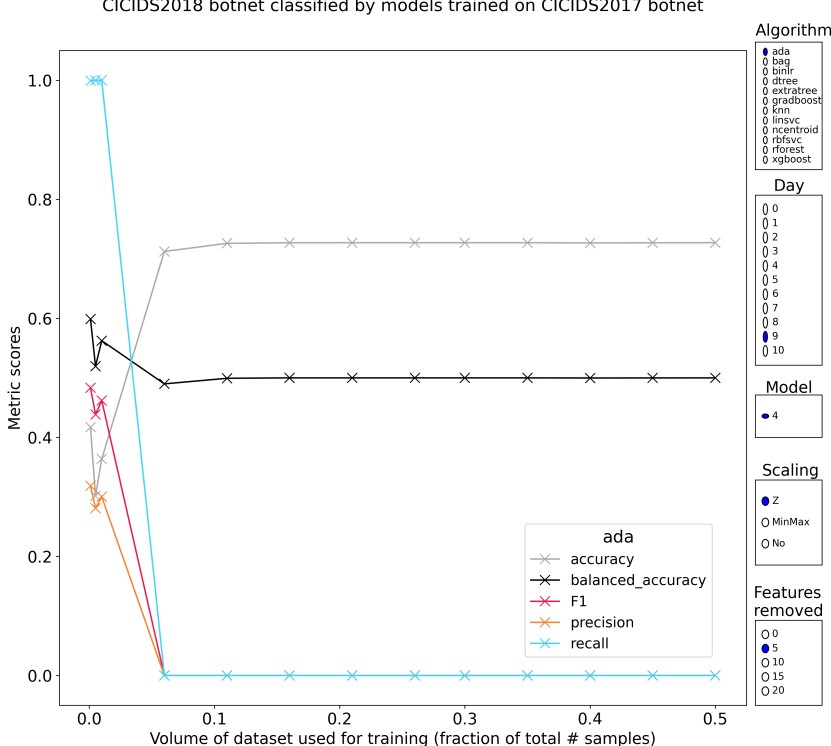

(**c**) Adaboost, pretrained for botnet, still normalized, top 5 features removed

**Figure 8.** The wild fluctuations between pretrained models when employing different scaling methods during preprocessing or when removing top-features when classifying a medium-difficulty class (botnet).

Nearest neighbors has an inverse relationship with training volume because precision–recall most often stays below 50% and it is not worth considering. The nearest centroids classifier has many good models that manage a balanced accuracy score of 75–80% and with very high stability. These models were trained with normalized features and have recall stable near 100%. Min–max-scaled features yield models with recall at a stable 50%. Not scaling features before training yields models with a stable recall of 0%. What is most interesting is that this happened at all considered points of training volume. Even within CIC-IDS2017, it was not advisable to use it as a classifier for malicious samples, due to its low precision. This remains unchanged on the botnet data of CSE-CIC-IDS2018.

The results of the logistic regression models and linear SVM are not good enough, but that was expected because these models performed poorly on the botnet data in CIC-IDS2017. The only interesting conclusion is the complete loss of recall on the botnet data in CSE-CIC-IDS2018, whereas these models trained to very high recall values on the 2017 data. Rbf-svc models have great performance (85% balanced accuracy, perfect recall with 55% precision) at very low training volumes (mostly with minmax-scaled features). This drops to approximately 70% balanced accuracy with increased training volume. These results are stable with regard to feature reduction. The loss in performance is mostly due to a sharp decline in recall.

### 5.2.9. Intermediate Conclusion

After the disappointing generalization strength of the global models, as discussed in Sections 4.2.4, a new hypothesis was formulated which states that models trained on specific attack classes might generalize better than their global two-class counterparts. This hypothesis is proven wrong by the results in the previous subsections (Sections 5.2.1–5.2.8). Model generalization rarely happens and when the pretrained models achieve stable, high classification metrics, it is most often on the easy classes of CIC-IDS2017.

There are three major issues that make the use of pretrained models so weak when it comes to generalization. First, how the features are scaled before training has a large impact on the model's performance, but there is no best choice that can be reasonably recommended. This was no issue for the models when they had to classify the test sets from CIC-IDS2017. Second, the relationship between training volume and classification scores is inverted more often than not, leading to a situation wherein models trained on 0.1–1% of the samples in CIC-IDS2017 perform best. The third and final nail in the coffin is the rapid loss in classification metrics when the most discriminative features are incrementally removed. Most models were very robust to this within CIC-IDS2017, especially to classify the easy classes, but this desirable property does not hold.

## 6. Discussion

The scope of this investigation has led to a substantial set of results. In the results (Sections 4 and 5), the classification results for all attack classes as well as the trends observed from the visualizations are described in detail. Even though both sections end with intermediate conclusions, they remain dense. This section was included to give a straightforward view of the results and their implications. Table 3 shows the best three models per attack class from both a baseline perspective (B rows) and a generalized performance perspective (G rows). What is best is determined by the ranking of the models based on an equally weighted combination of balanced accuracy, F1-score and use of training data (less is better). The numeric columns (except for reduction) are percentages with a maximum of 100.

Several points stand out in this table. First the baseline scores for all classes except infiltration (due to poor representation in CIC-IDS2017) are extremely high. Second and most importantly, pretrained models generalized well to the classes with clear network footprints such as bruteforce, L7-DoS, DDoS and botnet to some extent. These results are without any additional training and most often achieved by models that had little access to training data (% training at or below 1%, further broken down into one third training–two thirds validation). Third, although tree-based models typically have the highest baseline scores, the best generalizing models are not always tree-based. There are, however, more than enough tree-based models that do have great general performance (hence, they did not overfit) so it is possible. This is mainly interesting from a model interpretability perspective. Fourth and finally, the discrepancy between the baseline performance and the general performance for the web attacks would go unnoticed in most analyses, erroneously concluding that the models perform well on the class. This conclusion also applies to the global 2-class models.

One crucial remark about Table 3 is that it obfuscates whether the models had a stable, generalized performance. This is most often not the case (as shown in the detailed results in Section 5). Improvements that guarantee stable, general class-level models after training should be sought after. These improvements could come from changes in data preparation and model selection based on generalization potential (either during or after training) or algorithmic modifications that improve robustness.

**Table 3.** Classification metrics for the best 3 models per attack class, both for baseline (B) and generalized (G) classification, with the mention of the preprocessing parameters.

| B/G | Class | Algorithm | Balanced Acc. | F1 | Precision | Recall | Scaling | Reduction | % Train |
|-----|-------|-----------|---------------|-----|-----------|--------|---------|-----------|---------|
| B | 0.Bruteforce | gradboost | 99.84 | 99.06 | 98.40 | 99.73 | No | 0 | 0.5 |
|   |              | extratree | 99.63 | 99.24 | 99.21 | 99.28 | Z | 0 | 0.5 |
|   |              | extratree | 99.64 | 99.63 | 99.97 | 99.28 | Z | 0 | 1.0 |
| G | 0.Bruteforce | xgboost | 100 | 100 | 100 | 100 | No | 0 | 0.5 |
|   |              | xgboost | 99.97 | 99.95 | 99.90 | 100 | No | 0 | 1.0 |
|   |              | gradboost | 99.06 | 98.38 | 96.81 | 100 | MinMax | 0 | 0.1 |

**Table 3.** *Cont.*

| B/G | Class | Algorithm | Balanced Acc. | F1 | Precision | Recall | Scaling | Reduction | % Train |
|---|---|---|---|---|---|---|---|---|---|
| | | xgboost | 99.85 | 99.79 | 99.71 | 99.88 | No | 0 | 0.5 |
| B | 1.L7-DoS | xgboost | 99.85 | 99.77 | 99.66 | 99.89 | Z | 0 | 0.5 |
| | | xgboost | 99.84 | 99.76 | 99.61 | 99.91 | MinMax | 0 | 0.5 |
| | | linsvc | 97.80 | 82.59 | 71.57 | 97.63 | MinMax | 5 | 1.0 |
| G | 1.L7-DoS | linsvc | 97.98 | 84.93 | 75.14 | 97.65 | MinMax | 5 | 6.0 |
| | | linsvc | 97.75 | 81.63 | 70.10 | 97.70 | MinMax | 5 | 11.0 |
| | | rforest | 99.75 | 99.81 | 99.65 | 99.78 | Z | 20 | 1.0 |
| G | 2.L7-DoS (HeartBleed) | linsvc | 99.49 | 99.50 | 99.95 | 99.04 | MinMax | 0 | 0.5 |
| | | gradboost | 99.65 | 99.72 | 99.60 | 99.85 | MinMax | 10 | 1 |
| | | xgboost | 97.58 | 96.92 | 98.72 | 95.18 | MinMax | 0 | 1.0 |
| B | 2.Web Attacks | xgboost | 98.83 | 98.75 | 99.86 | 97.66 | MinMax | 0 | 6.0 |
| | | extratree | 98.89 | 98.27 | 98.75 | 97.80 | MinMax | 0 | 6.0 |
| | | gradboost | 64.23 | 43.37 | 91.15 | 28.45 | No | 10 | 0.5 |
| G | 5.Web Attacks | dtree | 77.74 | 38.69 | 29.69 | 55.52 | No | 0 | 11.0 |
| | | ada | 64.36 | 41.35 | 73.76 | 28.73 | No | 0 | 1.0 |
| | | extratree | 61.75 | 38.05 | 100 | 23.50 | MinMax | 5 | 0.5 |
| G | 6.Web Attacks | gradboost | 61.66 | 36.92 | 88.59 | 23.32 | No | 10 | 0.5 |
| | | xgboost | 61.66 | 36.82 | 87.42 | 23.32 | MinMax | 10 | 0.5 |
| | | dtree | 88.89 | 71.79 | 66.67 | 77.78 | MinMax | 10 | 11.0 |
| B | 3.Infiltration | xgboost | 93.06 | 91.18 | 96.88 | 86.11 | Z | 0 | 35.0 |
| | | extratree | 88.89 | 86.15 | 96.55 | 77.78 | MinMax | 15 | 26.0 |
| | | ncentroid | 50.19 | 17.50 | 11.33 | 38.42 | Z | 20 | 6.0 |
| G | 7.Infiltration | binlr | 50.46 | 14.86 | 11.71 | 20.34 | Z | 0 | 6.0 |
| | | binlr | 49.93 | 15.22 | 11.18 | 23.82 | Z | 20 | 6.0 |
| | | ncentroid | 57.75 | 42.42 | 35.74 | 52.16 | Z | 20 | 6.0 |
| G | 8.Infiltration | binlr | 55.23 | 43.21 | 31.58 | 68.38 | MinMax | 15 | 11.0 |
| | | linsvc | 51.09 | 43.11 | 28.62 | 87.29 | MinMax | 10 | 11.0 |
| | | xgboost | 98.42 | 98.19 | 99.58 | 96.85 | MinMax | 0 | 6.0 |
| B | 4.Botnet | xgboost | 98.14 | 97.65 | 99.06 | 96.29 | Z | 0 | 6.0 |
| | | xgboost | 97.53 | 97.32 | 99.68 | 95.07 | No | 0 | 6.0 |
| | | ada | 98.90 | 98.40 | 98.39 | 98.41 | No | 0 | 0.5 |
| G | 9.Botnet | gradboost | 92.11 | 82.67 | 70.51 | 99.91 | Z | 0 | 0.5 |
| | | ada | 92.11 | 82.67 | 70.51 | 99.91 | Z | 0 | 0.5 |
| | | extratree | 99.89 | 99.89 | 99.96 | 99.82 | Z | 5 | 0.1 |
| B | 5.DDoS | extratree | 99.87 | 99.88 | 99.96 | 99.79 | Z | 10 | 0.1 |
| | | extratree | 99.84 | 99.86 | 99.88 | 99.83 | No | 10 | 0.1 |
| | | dtree | 96.30 | 96.75 | 96.09 | 97.42 | Z | 0 | 0.5 |
| G | 3.DDoS | ada | 96.30 | 96.75 | 96.09 | 97.42 | Z | 0 | 1.0 |
| | | bag | 95.96 | 96.49 | 95.58 | 97.42 | Z | 0 | 0.5 |
| | | binlr | 99.86 | 99.87 | 99.99 | 99.75 | MinMax | 0 | 0.1 |
| G | 4.DDoS | binlr | 99.86 | 99.87 | 99.99 | 99.75 | MinMax | 5 | 0.1 |
| | | binlr | 99.86 | 99.87 | 99.99 | 99.75 | MinMax | 10 | 0.1 |
| | | xgboost | 99.82 | 99.73 | 99.75 | 99.70 | No | 0 | 0.5 |
| B | 7.Global | xgboost | 99.68 | 99.62 | 99.84 | 99.40 | Z | 0 | 0.5 |
| | | xgboost | 99.86 | 99.80 | 99.85 | 99.75 | Z | 0 | 1.0 |
| | | knn | 81.36 | 71.98 | 64.46 | 81.47 | Z | 0 | 0.1 |
| G | 10.Global | knn | 79.23 | 69.06 | 60.65 | 80.18 | Z | 10 | 1.0 |
| | | knn | 79.03 | 68.45 | 58.49 | 82.50 | Z | 10 | 0.5 |

## 7. Conclusions and Future Work

ML-based intrusion detection systems have to be able to accurately classify new samples to protect live networks. Getting access to these new samples can be tricky, but an intermediate evaluation is possible. This article tested whether a suite of supervised ML algorithms trained on CIC-IDS2017 (both global and class-specific models) effectively generalizes to the very similar, compatible CSE-CIC-IDS2018.

Unfortunately, our experiments demonstrated that the global, two-class models which had excellent performance on CIC-IDS2017 [11] do not generalize to the follow-up dataset CSE-CIC-IDS2018.

Even the most data-constrained trained models show clear signs of overfitting (best results at very low training volume) and an overall very weak performance. The best two-class models are the logistic regression and linear- and rbf-kernel SVMs. These reach between 90 and 100% recall with 50–60% precision. This leads to overall class separability in the 70% range. This is not sufficiently reliable to be used in real network defense systems.

Because the global models are too unreliable, specialized models for all shared attack classes between CIC-IDS2017 and CSE-CIC-IDS2018 have also been tested. Those results have pockets of good performance, mostly on the network-centric classes. Some models are able to classify the novel DoS, DDoS, botnet and brute force samples of CSE-CIC-IDS2018 with the retention of their strong performance metrics from classification within CIC-IDS2017 (F1-score > 95%). Section 6 provides a condensed version of the top results and their implications.

Three key issues still undermine a recommendation to use the tested algorithms in real network defense systems. First, how features are scaled has a major impact on the models' performance and the best choice varies too much to give a solid recommendation. This was no issue for the same models when only classifying the test sets of the data on which they were trained. Second, almost every model significantly struggles to maintain performance if a selection of top-features was removed prior to training. This too was much less of an issue for the models during standard intra-dataset testing. Third, the best-performing models were most often those trained on very little data (0.1–1% training volume). This clear sign of overfitting was most prominent for tree-based learners, but affected all other methods to some extent. Performance regressions by the numbers were erratic and could dip down to balanced accuracies of 50%.

Losing this invariance to scaling, training volume and feature reduction that made the models so attractive when classifying only within CIC-IDS2017 has a big implication. A large collection of models have to be trained and tested before the best models are cherry-picked. Such a large expenditure of time and computational resources for a relatively low yield is not defensible.

To summarize: this article experimentally demonstrates that ML-NIDS methods fail to generalize even just across tightly coupled datasets. Consequently, it is highly unlikely that they will perform well when deployed on real-world networks. We urge researchers in the ML-NIDS domain to execute this article's more rigorous model evaluation strategy to avoid publishing potentially misleading and overly optimistic results.

### *Future Work and Hypotheses*

Our future research will investigate potential solutions to improve ML-based NIDS systems until they can consistently classify related and compatible datasets.

An obvious first attempt would be to investigate more powerful classification methods. Recent ML-NIDS literature borrows neural network architectures that dominate pattern recognition tasks in other fields [37–39]. Although the results are great, no literature exists that tests whether they are better at generalization.

Alternatively, more stringent model regularization techniques and/or feature selection can be tested as potential solutions. Because the feature selection method of [11] was counter-intuitive, there is room to test optimized models that only kept the most potent features.

Finally, instead of trying to build global two-class models or attack-class specific models, the models could be trained to recognize attacks within specific domains (e.g., simultaneously training with differentiation at the protocol and attack level or training models to recognize traffic from a specific botnet). The major downside to this approach is that it reduces the range of attacks that it covers, thereby moving closer to signature-based methods.

**Author Contributions:** Conceptualization, L.D.; methodology, L.D.; software, L.D.; validation, L.D.; formal analysis, L.D.; investigation, L.D.; resources, L.D. and IDLab-Imec; data curation, L.D.; writing—original draft preparation, L.D.; writing—review and editing, L.D., M.V., T.W. and B.V.; visualization, L.D.; supervision, T.W., B.V. and F.D.T.; project administration, L.D.; funding acquisition, T.W., B.V. and F.D.T. All authors have read and agreed to the published version of the manuscript.

**Funding:** This research received no external funding.

**Institutional Review Board Statement:** Not applicable.

**Informed Consent Statement:** Not applicable.

**Data Availability Statement:** The datasets, both raw and cleaned up, are publicly available at https://gitlab.ilabt.imec.be/lpdhooge/ids-dataset-collection. The full source code of the analysis and the complete set of visualizations is available at https://gitlab.ilabt.imec.be/lpdhooge/reduced-unseen-testing. (both URLs checked on 10 March 2023).

**Conflicts of Interest:** The authors declare no conflict of interest.

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
