# Peer review of "Characterizing the Impact of Data-Damaged Models on Generalization Strength in Intrusion Detection"

_jcp, doi:10.3390/jcp3020008_

Round 1

Reviewer 1 Report

Concern 1#

The literature survey section must be expanded and the author is requested to cite the following latest paper related to this field 

Quadir, M. A., Christy Jackson, J., Prassanna, J., Sathyarajasekaran, K., Kumar, K., Sabireen, H., ... & Vijaya Kumar, V. (2020). An efficient algorithm to detect DDoS amplification attacks. Journal of Intelligent & Fuzzy Systems, 39(6), 8565-8572.

Concern 2#

The author must include the problem statement and must clearly say what problem the paper is trying to solve

Concern 3#

The author must clearly explain the flow of work through the proposed architectural diagram. The diagram must have contents related to the detailed process

Concern4#

The study's fundamental structure has to be explained with the help of an algorithm

Concern 5#

 How have the outcomes been ensured in light of the major uncertainties?

Concern6#

The conclusion has to be revised to incorporate the following advice: - Highlight your analysis and just present the most important takeaways from the full paper.

- Mention the advantages.

- In the final sentence of this section, mention the inference.

Make sure the topic of the Conclusion differs from what is discussed in the abstract.

- include the future work with multifaceted work

Author Response

Dear reviewer,   Thank you for reading our work and for outlining paths for its improvement. Below, I have included point-by-point responses to your comments.   1. The literature survey section must be expanded and the author is requested to cite the following latest paper related to this field Quadir, M. A., Christy Jackson, J., Prassanna, J., Sathyarajasekaran, K., Kumar, K., Sabireen, H., ... & Vijaya Kumar, V. (2020). An efficient algorithm to detect DDoS amplification attacks. Journal of Intelligent & Fuzzy Systems, 39(6), 8565-8572.   The literature section has been updated to include more appropriate, more recent work. It has also been restructured to improve the readability of the article.

2. The author must include the problem statement and must clearly say what problem the paper is trying to solve   An explicit problem statement subsection has been added (subsection 1.1) in which the core research idea and its novelty are discussed.

3. The author must clearly explain the flow of work through the proposed architectural diagram. The diagram must have contents related to the detailed process   A new graphic has been included that provides the necessary information to reproduce the experiment.

4. The study's fundamental structure has to be explained with the help of an algorithm   In our view, this ties in with the previous comment. The graphic, when read top-to-bottom is a visualized version of the experiment's algorithm. We have not duplicated this information in an additional LaTeX algorithmic block.

5. How have the outcomes been ensured in light of the major uncertainties?   The quality of the pretrained models has been validated as part of a previous article (https://ieeexplore.ieee.org/document/8901110). Review of the cross-validation scores confirmed the excellent classification results and their stability. With regard to the problem of poor inter-dataset generalization, we have continued to investigate this. The problem also exists when pretraining on CSE-CIC-IDS2018 and classifying CIC-IDS2017. Furthermore, poor inter-dataset generalization is not solved by using more advanced classifiers. We have tested various other models including state-of-the-art neural-network-based IDS and they equally lose significant portions of their classifying ability. The problem of weak generalization is also not solved by using models with only the best features. We aim to publish a summary article with our attempts to solve this fundamental ML-NIDS problem later in 2023.   6. The conclusion has to be revised to incorporate the following advice: - Highlight your analysis and just present the most important takeaways from the full paper. - Mention the advantages. - In the final sentence of this section, mention the inference. - Make sure the topic of the Conclusion differs from what is discussed in the abstract. - include the future work with multifaceted work   The conclusion has been reworked. It is now shorter and more focused. The most important numerical results remain as do the three key issues discovered during inter-dataset generalization testing of ML-NIDS models. The final paragraph of the conclusion spells out its central recommendation and advantage to other ML-NIDS researchers. The future work has shortened and partially rewritten to be more aligned with the article. It now clearly states three different approaches to try to solve generalization for machine-learned intrusion detection systems.  

Reviewer 2 Report

All figure should be explained clearly and can be redrawn to make fonts visible.

The novelty of the work proposed in the paper needs to be discussed.

The quality of the tables is poor. 

Author Response

Dear reviewer,
Thank you for reading our work and for outlining paths for its improvement. Below, I have included point-by-point responses to your comments.
1. All figure should be explained clearly and can be redrawn to make fonts visible.
All figures have been improved to be more legible. They are now formatted in compliance with the MDPI camera-ready style set out by the editorial office.   Every figure is referenced in the text with additional context and each figure has an appropriate caption.
2. The novelty of the work proposed in the paper needs to be discussed.
The novelty of the work is now clearly discussed in subsection 1.2 (research contribution).
3. The quality of the tables is poor.
The quality of all tables has been improved by following the MDPI camera-ready style set out by the editorial office.

Reviewer 3 Report

The authors have done a good work. However, please find the following concerns.

1- Most of the referred research/refs are old; the authors need to refer to the important up to date works and show how their work can contribute the body of the knowledge and superior others' works.

2- In the contribution section, it has been claimed that there is a formal analysis, but actually no explicit formal analysis is found. Please show this clearly.

3- The authors need to re-check the correctness of the punctuation marks, especially in the abstract.

4-The authors need to re-check the Capitalization cases, as example, in some figures, the figure title has been capitalized with each word and others were not.

Author Response

Dear reviewer,
Thank you for reading our work and for outlining paths for its improvement. Below, I have included point-by-point responses to your comments.
1. Most of the referred research/refs are old; the authors need to refer to the important up to date works and show how their work can contribute the body of the knowledge and superior others' works.
The literature section has been updated to include more appropriate, more recent work. It has also been restructured to improve the readability of the article.
2. In the contribution section, it has been claimed that there is a formal analysis, but actually no explicit formal analysis is found. Please show this clearly.
Our apologies, this is a miscommunication. We have followed Elsevier's CRediT author statement (https://www.elsevier.com/authors/policies-and-guidelines/credit-author-statement) which includes formal analysis as a category defined as "Application of statistical, mathematical, computational, or other formal techniques to analyze or synthesize study data". As the article centers around the shortcomings of machine-learned intrusion detection systems under more challenging evaluation, we included the category because it falls within the aforementioned definition. For now, to maintain a full CRediT author statement, we have kept the category, but it can be removed if necessary.
3. The authors need to re-check the correctness of the punctuation marks, especially in the abstract.
The abstract has been reviewed and edited with special attention for punctuation marks.
4. The authors need to re-check the Capitalization cases, as example, in some figures, the figure title has been capitalized with each word and others were not.
The article has been reviewed and edited for capitalization consistency, including the figures and tables.

Reviewer 4 Report

This article aims to investigate the capability of machine learning methods in the context of learning meaningful representations of network attacks.

The article is well-written and easy to follow. Both (in the abstract and Introduction), the problem statement and its importance are nicely articulated. Moreover, the related work (section 1.1) analyzes state-of-the-art in the context of the formulated research problem. However, the Introduction section generally provides a brief description of the proposed method (investigation methodology in this article), contributions, and significance of achieved results. Therefore, the addition of this missing information in the Introduction may enhance the impact of the article.

Materials and methods (Section 2)  is very well written. The target data sets (such as )CIC-IDS2017, CSE-CIC-IDS2018) are briefly summarized. Furthermore, the training & evaluation procedure is presented in Section 2.2. In this section, it is better to address the following two issues: (a) the motivation behind the selected datasets and algorithms (b) A structural representation of the proposed method. Training and evaluation procedure may just be part of the proposed investigation methodology. The description of this methodology is important. Can this methodology be applied to algorithms data sets and applications  (not discussed in this paper). How much general purpose the proposed methodology is ??

The obtained results are significant. The experimentation methodology is sound. The results are presented in a very nice way.

To summarize, the article addresses an interesting problem and performs technically sound research with a nice discussion on the achieved results.

Author Response

Dear reviewer,
Thank you for reading our work and for outlining detailed paths for its improvement. Below, I have included point-by-point responses to your comments.
This article aims to investigate the capability of machine learning methods in the context of learning meaningful representations of network attacks.
1. The article is well-written and easy to follow. Both (in the abstract and Introduction), the problem statement and its importance are nicely articulated. Moreover, the related work (section 1.1) analyzes state-of-the-art in the context of the formulated research problem. However, the Introduction section generally provides a brief description of the proposed method (investigation methodology in this article), contributions, and significance of achieved results. Therefore, the addition of this missing information in the Introduction may enhance the impact of the article.
The introduction is now more clear about the motivation for the article, its methodology and its contributions. The significance is mentioned in the abstract and as the final paragraph of the conclusion. Proposed ML-IDS systems that only evaluate intra-dataset likely overstate their general effectiveness so researchers should adopt a more rigorous model evaluation.  
2. Materials and methods (Section 2)  is very well written. The target data sets (such as )CIC-IDS2017, CSE-CIC-IDS2018) are briefly summarized. Furthermore, the training & evaluation procedure is presented in Section 2.2. In this section, it is better to address the following two issues: (a) the motivation behind the selected datasets and algorithms (b) A structural representation of the proposed method. Training and evaluation procedure may just be part of the proposed investigation methodology. The description of this methodology is important. Can this methodology be applied to algorithms data sets and applications  (not discussed in this paper). How much general purpose the proposed methodology is ??
The obtained results are significant. The experimentation methodology is sound. The results are presented in a very nice way.
To summarize, the article addresses an interesting problem and performs technically sound research with a nice discussion on the achieved results.

The motivation for the two CIC-IDS datasets is now made explicit as part of the data set description. The methodology is improved by a new graphic that summarizes the structure of the experiment (to accompany section 2.2 Training & Evaluation Procedure).
The methdology is general purpose, but there are a few requirements. Most importantly, the datasets need a shared feature set extracted in the same way (by the same toolchain, with the same configuration). The preprocessing of the data also needs to be identical and it is necessary that the represented attacks are similar (or at least fall within the same attack category).
Testing generalization in this way has been done in computer vision and natural language processing research with surprising results. Typically, it is easier to obtain / create unseen, but related data to do generalization testing. The landmark article in computer vision regarding this topic is: 'Do ImageNet Classifiers Generalize to ImageNet' presented at ICML2019 in which the authors recreate the data generation for several benchmark datasets to generate new test sets and re-evaluate state-of-the-art models (http://proceedings.mlr.press/v97/recht19a/recht19a.pdf)

Round 2

Reviewer 2 Report

There should be links between paragraphs and sections which should lead to the next one.

Author Response

Dear reviewer,

Thank you for providing an additional comment to improve the article.

1. There should be links between paragraphs and sections which should lead to the next one.

We have added much more linking, both forward and backward, primarily to the literature, materials & methods and result sections. The additional summaries with clickable links should give readers an easier overview and a quicker method to move to the sections which interest them the most.